# Thromboelastometry demonstrates endogenous coagulation activation in nonsevere and severe COVID-19 patients and has applicability as a decision algorithm for intervention

**Rodrigo B. Aires**[1,2☯*], **Alexandre A. de S. M. Soares**[1], **Ana Paula M. Gomides**[3], **André M. Nicola**[1], **Andréa Teixeira-Carvalho**[4], **Dayde Lane M. da Silva**[5], **Eliana T. de Gois**[1,6], **Flávia D. Xavier**[1], **Francielle P. Martins**[1,7], **Gabriela P. J. Santos**[1,8], **Heidi Luise Schulte**[1], **Isabelle S. Luz**[9], **Laila S. Espindola**[1,10], **Laurence R. do Amaral**[11], **Liza F. Felicori**[12], **Luciana A. Naves**[1,13], **Maíra R. M. de Carvalho**[1,14], **Matheus de S. Gomes**[11], **Otávio T. Nóbrega**[1], **Patrícia Albuquerque**[15], **Wagner Fontes**[9], **Ciro M. Gomes**[1,16], **Patricia S. Kurizky**[1,16☯*], **Cleandro P. Albuquerque**[1,17], **Olindo A. Martins-Filho**[4‡], **Licia Maria H. da Mota**[1,17‡]

**1** Programa de Pós-graduação em Ciências Médicas da Faculdade de Medicina da Universidade de Brasília (UnB), Brasília, Distrito Federal, Brazil, **2** Serviço de Anestesiologia, Hospital Santa Lúcia Sul, Brasília, Distrito Federal, Brazil, **3** Faculdade de Ciências da Saúde, Centro Universitário de Brasília, Brasília, Distrito Federal, Brazil, **4** Grupo Integrado de Pesquisa em Biomarcadores, Fundação Oswaldo Cruz (FIOCRUZ), Belo Horizonte, Minas Gerais, Brazil, **5** Departamento de Farmácia, Universidade de Brasília (UnB), Brasília, Distrito Federal, Brazil, **6** Unidade de Clínica Médica, Hospital Regional do Gama (HRG), Brasília, Distrito Federal, Brazil, **7** Núcleo de Gestão Interna de Leitos, Hospital Regional da Asa Norte (HRAN), Brasília, Distrito Federal, Brazil, **8** Unidade de Medicina Interna, Hospital Regional de Taguatinga, Brasília, Distrito Federal, Brazil, **9** Laboratório de Bioquímica e Química de Proteínas, Departamento de Biologia Celular, Instituto de Ciências Biológicas, Universidade de Brasília (UnB), Brasília, Distrito Federal, Brazil, **10** Laboratório de Farmacognosia, Faculdade de Ciências da Saúde, Universidade de Brasília (UnB), Brasília, Distrito Federal, Brazil, **11** Laboratório de Bioinformática e Análises Moleculares, Rede Multidisciplinar de Pesquisa, Ciência e Tecnologia (RMPCT), Universidade Federal de Uberlândia, Patos de Minas, Minas Gerais, Brazil, **12** Laboratório de Biologia Sintética e Biomiméticos, Departamento de Bioquímica e Imunologia, Instituto de Ciências Biológicas, Universidade Federal de Minas Gerais (UFMG), Belo Horizonte, Minas Gerais, Brazil, **13** Serviço de Endocrinologia, Hospital Universitário de Brasília, Universidade de Brasília (UnB), Brasília, Distrito Federal, Brazil, **14** Unidade de Medicina Interna, Hospital Regional da Asa Norte (HRAN), Brasília, Distrito Federal, Brazil, **15** Faculdade UnB Ceilândia, Universidade de Brasília (UnB), Brasília, Distrito Federal, Brazil, **16** Serviço de Dermatologia, Hospital Universitário de Brasília, Universidade de Brasília (UnB), Brasília, Distrito Federal, Brazil, **17** Serviço de Reumatologia, Hospital Universitário de Brasília, Universidade de Brasília (UnB), Brasília, Distrito Federal, Brazil

☯ These authors contributed equally to this work.
‡ OAMF and LMHM also contributed equally to this work.
* airesrod@gmail.com (RBA); patyshu79@gmail.com (PSK)

## Abstract

In patients with severe forms of COVID-19, thromboelastometry has been reported to display a hypercoagulant pattern. However, an algorithm to differentiate severe COVID-19 patients from nonsevere patients and healthy controls based on thromboelastometry parameters has not been developed. Forty-one patients over 18 years of age with positive qRT-PCR for SARS-CoV-2 were classified according to the severity of the disease: nonsevere (NS, n = 20) or severe (S, n = 21). A healthy control (HC, n = 9) group was also

**Data Availability Statement:** All relevant data are within the manuscript and its Supporting Information files.

**Funding:** The authors are grateful for the financial support provided by Conselho Nacional de Densenvolvimento Científico e Tecnológico (CNPq). LFF thank the financial support provided by Conselho Nacional de Desenvolvimento Científico e Tecnológico (CNPq), Coordenação de Aperfeiçoamento de Pessoal de Nível Superior (CAPES), under process no. 88887.506611/2020-00 and 88887.504420/2020-00 and Fundação de Amparo à Pesquisa de Minas Gerais (FAPEMIG), under process no. REDE-00140-16. LSE thank the ArboControl Brazil Project funded by the Ministry of Health, under process no. TED 74/2016 & TED 42/2017 for financial support. LAN, MSG, OTN, WF and OAMF thank CNPq for the PQ fellowship program. The funders had no role in the study design, data collection and analysis, decision to publish, or preparation of the manuscript.

**Competing interests:** The authors have declared that no competing interests exist.

examined. Blood samples from all participants were tested by extrinsic (EXTEM), intrinsic (INTEM), non-activated (NATEM) and functional assessment of fibrinogen (FIBTEM) assays of thromboelastometry. The thrombodynamic potential index (TPI) was also calculated. Severe COVID-19 patients exhibited a thromboelastometry profile with clear hypercoagulability, which was significantly different from the NS and HC groups. Nonsevere COVID-19 cases showed a trend to thrombotic pole. The NATEM test suggested that nonsevere and severe COVID-19 patients presented endogenous coagulation activation (reduced clotting time and clot formation time). TPI data were significantly different between the NS and S groups. The maximum clot firmness profile obtained by FIBTEM showed moderate/elevated accuracy to differentiate severe patients from NS and HC. A decision tree algorithm based on the FIBTEM-MCF profile was proposed to differentiate S from HC and NS. Thromboelastometric parameters are a useful tool to differentiate the coagulation profile of nonsevere and severe COVID-19 patients for therapeutic intervention purposes.

## Introduction

Patients with coronavirus disease 2019 (COVID-19) caused by severe acute respiratory syndrome coronavirus-2 (SARS-CoV-2) have shown an increased frequency of thromboembolic phenomena since the beginning of the pandemic, which represents a high morbidity-mortality burden [1–3]. The pathophysiology of these findings is not fully established, although information available to date shows that the changes in the hemostasis system seem to be triggered by the high production of proinflammatory cytokines [1–5]. The massive production of IL-1β, IL-6 and TNF-α, among others, leads to a parallel increase in fibrinolysis inhibitors, such as plasminogen activator inhibitor (PAI-1) and thrombin activated fibrinolysis inhibitor (TAFI), increased expression of tissue factor by circulating mononuclear cells and release of neutrophil extracellular traps (NETs) [1, 4–6].

The standard tests for the evaluation of coagulopathies are activated partial thromboplastin time (APTT), prothrombin time and activity (PTA), d-dimer and platelet count [4, 7]. However, these methods present several limitations because they only cover the initial phase of coagulation and do not evaluate the different components involved in the dynamics of clot formation. In this sense, viscoelastic methods traditionally used to monitor hemorrhagic disturbances have emerged as a possible tool for assessing the hemostasis profile in the thrombotic pole of blood coagulation disorders [8, 9]. Indeed, reduced clotting time (CT) and increased maximum clot firmness (MCF) have been used to characterize hypercoagulability conditions [10–12]. In this scenario, the thrombodynamic potential index (TPI) [10] constitutes an alternative parameter for monitoring the risk of thromboembolic events by representing the global coagulation process [8, 9, 13].

Few thromboelastometry studies have evaluated the coagulopathic process in COVID-19, with most being performed in critically ill patients, where a prothrombotic profile is characterized by reduced CT and CFT along with increased MCF of functional fibrinogen (FIBTEM), extrinsic (EXTEM) and intrinsic (INTEM) assays [14–16]. To date, whether nonsevere patients also develop coagulation derangements, that could increase their risk of thrombotic complications, remains undetermined. Even in severe patients, the effects of pathophysiologic alterations on the initial stages of hemostasis have not been determined. In this regard, the use of nonactivated temogram (NATEM) could be used as a complementary method to evaluate

the participation of circulating tissue factor, which has been observed in patients with bacterial sepsis [17–20].

Thromboelastometry could be a useful tool for better assessing the coagulation profile of nonsevere and severe patients, which would help clinicians choose the most appropriate thromboprophylaxis intervention. The aim of this study was to characterize the coagulation process in nonsevere and severe forms of COVID-19 compared to that of healthy controls.

Our study was able to demonstrate that even nonsevere patients already show coagulation derangements.

## Patients and methods

### Experimental design

The present study is part of a major investigation protocol named the TARGET project (http://dx.doi.org/10.2196/24211): a longitudinal observational study carried out at tertiary hospitals responsible for the care of COVID-19 patients during the pandemic in midwestern Brazil (Hospital Regional da Asa Norte and Hospital Universitário de Brasília, Brasília, DF, Brazil) [21]. This study was registered on the Brazilian Registry of Clinical Trials Platform (ReBEC, RBR-62zdkk) and approved by the National Commission for Ethics in Research in Brazil (CONEP, CAAE 30846920.7.0000.0008). STROBE recommendations for observational studies were followed.

In order to avoid unnecessary manipulation of objects between healthy and sick individuals in a pandemic situation, all study participants signed an electronic informed consent, which was approved by our institutional review board.

### Study population

Forty-one patients over 18 years of age with positive qRT-PCR for SARS-CoV-2 were recruited and classified into two groups according to the severity of the disease. The nonsevere (NS, n = 20) group included patients with no need for hospitalization, and the severe (S, n = 21) group included patients with a need for hospital care due to [22]:

1. dyspnea (respiratory rate >30 respiratory incursions per minute), S

2. SpO2 <93% in room air,

3. PaO2/FiO2 <300 mmHg,

4. admission to the intensive care unit (ICU) or

5. need for mechanical ventilation.

Nine healthy controls (HCs, n = 9) with SARS-CoV-2-negative qRT-PCR from 5 to 7 days before blood collection were enrolled to establish local thromboelastometric reference values for ROTEM. The demographic parameters (age and sex) as well as the clinical records (weight, body mass index, comorbidities, use of angiotensin receptor blockers, symptoms, chest CT, ICU admission and treatments prescribed) are detailed in Table 1.

Peripheral venous blood (4 mL) was collected from each participant in vacuum tubes, with 3.8% citrate used as an anticoagulant. Blood sampling of COVID-19 patients was carried out from 7 to 10 days after the diagnostic confirmation of the disease. The hemostasis assessment included only thromboelastometric tests in a single evaluation. All tests were performed within 4 hours after blood sampling.

**Table 1. Baseline characteristics of the study participants.**

| Parameter* | Healthy Control | Non-Severe | Severe |
|---|---|---|---|
| | HC, n = 09 | NS, n = 20 | S, n = 21 |
| Age, Median, Min-Max | 40, 23–51 | 39, 19–70 | 50, 26–76 |
| Gender, F/M (%) | 9/0 (100) | 8/12 (40) | 12/9 (57) |
| Weight, kg, mean±SD | 74±8 | 78±19 | 90±16 |
| BMI † | | | |
| ≤24.9 | 3 (33) | 6 (30) | 2 (9) |
| 25–29.9 | 3 (33) | 11 (55) | 6 (29) |
| ≥30 | 3 (33) | 3 (15) | 13 (62) |
| Hypertension | 0 (0) | 0 (0) | 10 (48) |
| Obesity | 3 (33) | 3 (15) | 13 (62) |
| Use of ARB | 0 (0) | 4 (20) | 9 (43) |
| Symptoms (%) ‡ | — | Anosmia 12 (63) | |
| | | Ageusia 12 (63) | Dyspnoea 17 (81) |
| | | Asthenia 12 (63) | Cough 14 (67) |
| | | Headache 12 (63) | Asthenia 12 (57) |
| | | Cough 11 (58) | Fever 11 (52) |
| Chest CT (%) § | | | |
| <25% | 0 (0) | 3 (15) | 6 (29) |
| 25–50% | 0 (0) | 0 (0) | 8 (38) |
| >50% | 0 (0) | 0 (0) | 7 (33) |
| ICU admission (%) | 0 (0) | 0 (0) | 5 (24) |
| Treatment (%) | — | Azythromycin 14 (70) | Corticoid 21 (100) |
| | | Corticoid 12 (60) | Azithromycin 20 (95) |
| | | Enoxaparin 4 (20) | Enoxaparin 19 (90) |

* Data are reported as number of subjects (%), except for age (median–Min-Max), sex (F/M, %) and weight (kg, mean±SD

† BMI, Body Mass Index, calculated as the ration between weight in kilograms by the square of the height in meters, considering ≤24.9 as healthy weight, 25–29.9 as overweight and ≥30 as obese. ARB, Angiotensin Receptor Blockers.

‡ Only symptoms that were present in more than 50% of the subjects were listed. Only the most frequent treatments offered to the subjects were listed.

§ Chest CT = Computed Tomography, considering the radiologist subjective evaluation of pulmonary parenchyma compromise (ground glass opacity), categorized as: 25%, 25–50% and >50%. ICU, Intensive Care Unit.

## Inclusion criteria

COVID-19 patients treated during the pandemic in midwestern Brazil (Hospital Regional da Asa Norte and Hospital Universitário de Brasília, Brasilia, DF, Brazil (between August 1st and September 30th, 2020) were included. Confirmatory diagnosis was based on positive SARS-CoV-2 infection results in oropharyngeal swabs by quantitative real-time polymerase chain reaction (qRT-PCR).

## Exclusion criteria

The exclusion criteria were as follows:

1. under 18 years old.

2. pregnancy.

3. thrombophilia or previous thromboembolic events.

4. previous use of anticoagulants.

5. previous use of antiplatelet drugs

6. surgical procedures in the last 4 weeks.

7. hereditary coagulopathies and

8. psychiatric diseases that impaired the understanding of the informed consent form.

## Thromboelastometry

Briefly, the ROTEMⓇ Delta device (Werfen, Barcelona, Spain) was used to perform the thromboelastometric analysis following the manufacturer's instructions (**protocol available:** dx.doi.org/10.17504/protocols.io.bwvbpe2n).

All tests were performed by the same experienced laboratory technician at the DASA clinical laboratory, Brasília, Brazil. The EXTEM, INTEM and NATEM outputs were analyzed to yield the following parameters:

1. CT = clotting time, expressed in seconds.

2. ALPHA = alpha angle, expressed in $^{\circ}$;

3. CFT = clot formation time, expressed in seconds.

4. MCF = maximum clot firmness, expressed in mm; and

5. ML = maximum lysis, expressed in %.

6. The thrombodynamic potential index (TPI) was calculated as $[(100 \times MCF)/(100-MCF)]/CFT$ [10]. In the FIBTEM assay, only MCF was considered, provided that the main function of this test is to analyze the participation of fibrinogen in clot firmness. An illustrative overview of the thromboelastometry principles and parameters measured in the temograms is displayed in Fig 1.

## Statistical analysis

The sample size calculation was carried out based on a previous study of NATEM curves in septic patients and healthy controls [23]. The G*Power software version 3.1.9.6 was used and, considering a power of 95%, effect size d = 1.664 and a maximum type I error of 5%, yielded a minimal sample of 11 patients in each group [24]. No data was excluded.

GraphPad Prism version 8.0.0 for Windows software (GraphPad Software, San Diego, California USA, www.graphpad.com) was used for the descriptive statistical analysis. A Normal distribution was tested with the Shapiro-Wilk test. Categorical variables were described as absolute and relative frequencies and analyzed with the Chi-square or Fisher exact test. Continuous variables were described as the mean ± SD. Multiple comparisons among groups were performed using one-way ANOVA followed by Tukey's test for pairwise comparisons. Two-tailed Student's t test was used for comparisons between nonsevere and severe patients. In all cases, p values < 0.05 indicated statistical significance.

ROC curves were constructed using MedCalc software, Version 7.3.0.0 (Ostend, Belgium, URL https://www.medcalc.org/), to define the cutoff values and estimate the global accuracy based on the area under the ROC curve (AUC). Performance indices expressed as percentages (sensitivity and specificity) were obtained for each thromboelastometric parameter in all ROTEM tests. TG-ROC curves were assembled to confirm the selected cutoffs.

Decision trees were built using WEKA software (Waikato Environment for Knowledge Analysis, version 3.6.11, University of Waikato, New Zealand, URL https://www.cs.waika to.

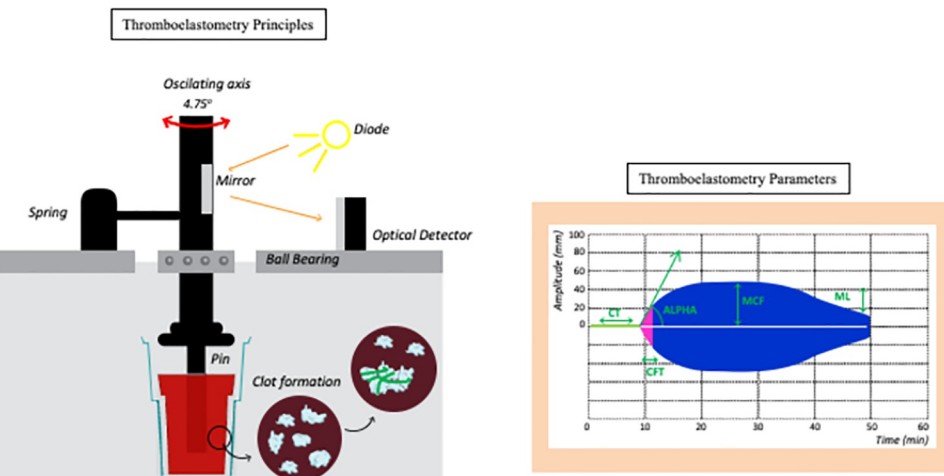

**Fig 1. Thromboelastometry method for clot evaluation.** A pin that spins around its own axis is put in contact with a citrated blood sample inside a cuvette. After recalcification and addition of a specific activator (depending upon the test), the clotting starts, and as it becomes firmer, the spinning capacity of the axis is reduced, which is transformed by the system in a graphic representation of the clot, with increasing amplitude. As fibrinolysis starts, the clot becomes less firm, which is represented as a decreasing amplitude on the monitor. EXTEM: thromboplastin is the activator, and it evaluates the extrinsic activation of coagulation; INTEM: elagic acid is the activator, and it evaluates the intrinsic activation of coagulation; FIBTEM: thromboplastin and cytochalasin D (which inhibits platelet activity) are added, and it only depicts the participation of fibrinogen in the clot; and NATEM: recalcified blood with no activator, it performs a nonactivated evaluation of coagulation. Circulating tissue factors, such as those expressed on monocytes in inflammatory states, will start the coagulation process. CT represents the clotting time (expressed in seconds), which is the timeframe from activation until an amplitude of 2 mm, and indicates thrombin formation; ALPHA (expressed in °) is defined as the angle between the middle axis and the tangent to the clotting curve through the 2 mm amplitude point and represents the dynamic polymerization of fibrin; CFT represents clot formation time(expressed in seconds) and indicates the dynamic polymerization of fibrin, and it is the timeframe between 2 mm and 20 mm of clot amplitude; MCF represents maximum clot firmness (expressed in mm), and it indicates the maximum amplitude of the clot and represents its main constituents, namely, fibrinogen and platelets; ML represents maximum lysis (expressed in %), and it indicates the percentage of clot reduction after initiation of fibrinolysis. Therefore, thromboelastometry analyzed 60 minutes after initiation depicts important information about every phase of the coagulation process.

ac.nz/ml/weka/) to classify COVID patients and healthy controls based on selected thromboelastometric parameters. Leave-one-out cross-validation (LOOCV) was applied to estimate the classification accuracy and test the generalizability of the model.

The graphics (bar diagrams, scatter charts and decision trees) were generated using Microsoft Office Package version 2012 and GraphPad Prism software, Version 8.0.

## Results

### Population characteristics

Age and sex distributions were similar between the nonsevere and severe groups (p = 0.1683 and p = 0.354, respectively). Severe patients showed significantly higher weight (21% higher) than HCs. For BMI≥30, the severe forms differed from the nonsevere forms (62% *vs* 15%). The percentage of hypertensive patients was higher in the severe than the nonsevere patients (48% *vs* 0%). Dyspnea, cough, and asthenia were the most frequent symptoms in severe patients, while anosmia, ageusia and asthenia were more frequent symptoms in nonsevere patients (Table 1).

### Thromboelastometry parameters in COVID-19 patients and healthy controls

A comparison of the thromboelastometric profiles of COVID-19 patients (NS+S) and healthy controls (HC) is shown in Fig 2. The data analysis demonstrated a significant increase in CT

values from the EXTEM temogram of COVID-19 patients compared to HCs. Moreover, COVID-19 patients exhibited significantly shorter CT values and higher ALPHA angle, MCF and TPI results from the NATEM test in comparison with healthy controls. Higher MCF values from FIBTEM were also reported for COVID-19 patients compared with HCs. Regardless of the thromboelastometric assay, differences were not observed for CFT or ML. Moreover, the INTEM parameters did not differ between the COVID-19 patients and HCs (Fig 2). A detailed description of the viscoelastic tests is provided in S1 Table.

## Thromboelastometry parameters in COVID-19 patients according to the disease severity

A comparative analysis of the temograms from COVID-19 patients with nonsevere or severe forms of the disease is presented in Fig 3. The data analysis demonstrated that patients with severe disease exhibited EXTEM results characterized by increased CT, ALPHA angle and MCF values and reduced CFT values compared to nonsevere patients and healthy controls. Notably, higher TPI values from EXTEM were observed in severe patients. No difference in the ML parameter was observed among the NS, S and HC patients (Fig 3).

An analysis of the INTEM thermogram in patients with severe disease, indicated a hypercoagulability profile with an increased ALPHA angle and MCF along with reduced CFT but unaltered CT and ML compared to the nonsevere patients and healthy controls. The TPI values from the INTEM assay were also increased in the S patients compared to the NS and HC patients (Fig 3).

Data from the NATEM thromboelastogram showed that regardless of disease severity, CT was significantly lower in COVID-19 patients than in HCs. However, the CFT value was lower in the S patients than the NS and HC patients. The ALPHA angle and MCF values were higher in the S group than in the NS and HC groups. In the NATEM test, the TPI values were higher in the S group than in the NS and HC groups (Fig 3).

The FIBTEM test showed a clear elevation of MCF values in severe patients relative to the NS and HC groups (Fig 3).

## Performance of thromboelastometry parameters as complementary biomarkers to classify COVID-19 patients

To further explore the applicability of thromboelastometry parameters from EXTEM, INTEM, NATEM and FIBTEM to cluster COVID-19 patients from healthy controls as well as subgroups of COVID-19 patients according to disease severity, the global accuracy (AUC) of each parameter was evaluated along with other performance indices (sensibility, specificity, negative and positive likelihood ratios) obtained from the ROC curve attributes. The results are presented in Tables 2 and 3.

A panoramic overview analysis was carried out based on AUC values higher than 0.7 as an indicator of moderate or elevated global accuracy. Based on these criteria, thirteen parameters were preselected for further analysis to classify COVID-19 *vs* HC: CT, ALPHA angle, CFT, MCF and TPI from EXTEM; MCF and TPI from INTEM; CT, ALPHA angle, CFT, MCF and TPI from NATEM and MCF from FIBTEM (Table 2). Additionally, thirteen attributes were used to cluster NS from S with moderate or elevated global accuracy (AUC>0.7): including ALPHA angle, CFT, MCF and TPI from EXTEM; ALPHA angle, CFT, MCF and TPI from INTEM; ALPHA angle, CFT, MCF and TPI from NATEM and MCF from FIBTEM (Table 2).

Further comparisons between the HC and NS patients indicated that three parameters exhibited moderate or elevated global accuracy (AUC>0.7): CT from EXTEM and CT and MCF from NATEM (Table 3). Additionally, sixteen parameters showed the ability to cluster

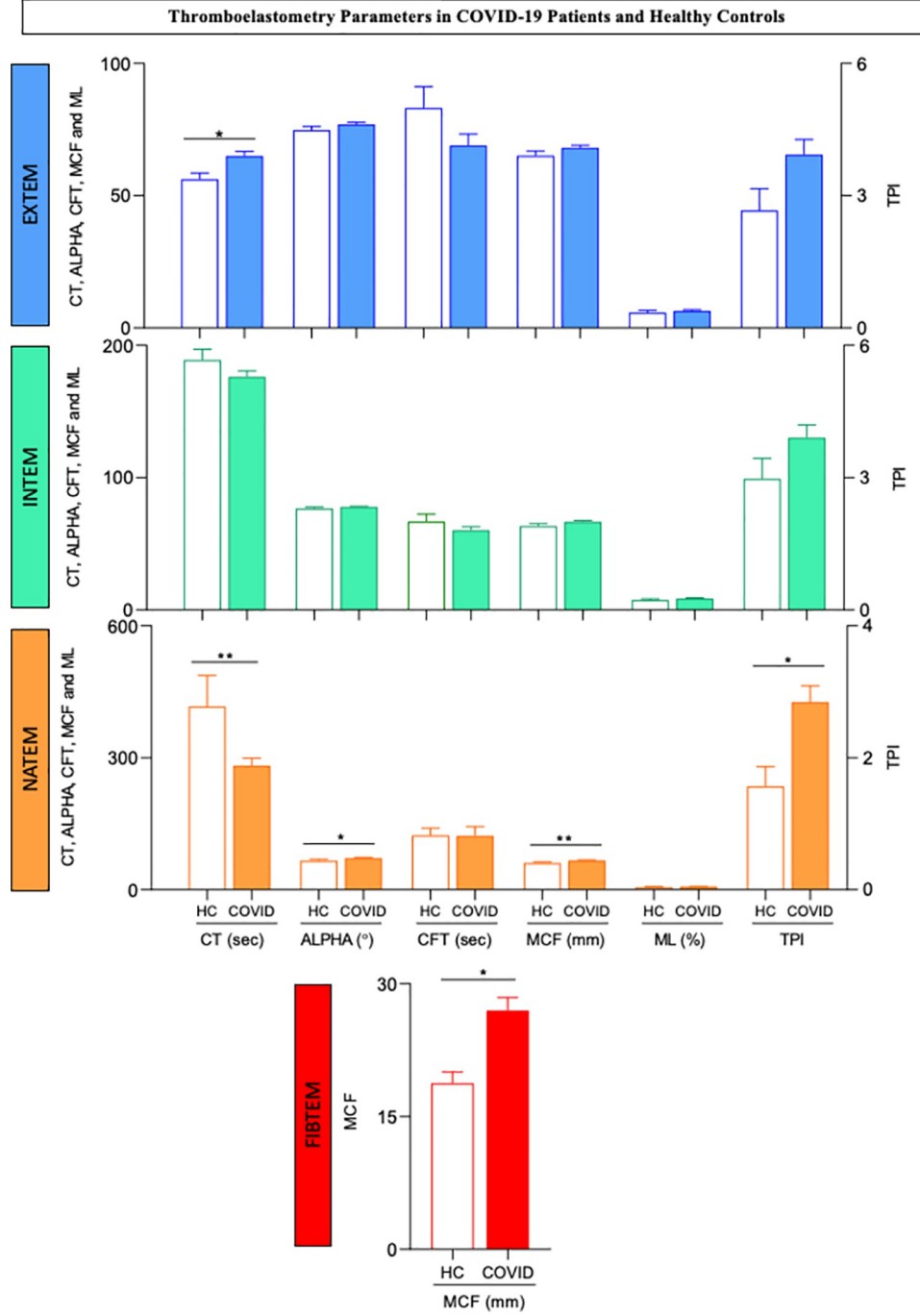

**Fig 2. Thromboelastometry parameters in COVID-19 patients and healthy controls.** Extrinsic (EXTEM) and intrinsic (INTEM) coagulation activity assays, nonactivated coagulation assays (NATEM) and functional assessments of fibrinogen assays (FIBTEM) were carried out as described in the Materials and Methods. The results for COVID-19 patients (COVID, n = 41, color bars) and healthy controls (HC, n = 9, white bars) are presented as the mean values ± standard error. CT = clotting time, expressed in seconds; ALPHA = alpha angle, expressed in °; CFT = clot formation time, expressed in seconds; MCF = maximum clot firmness, expressed in mm; ML = maximum lysis, expressed in % and TPI = thrombodynamic potential index, calculated as [(100 x MCF)/(100-MCF)]/CFT. Significant differences are highlighted by connecting lines and * or ** for p values ≤0.05 and ≤0.01, respectively.

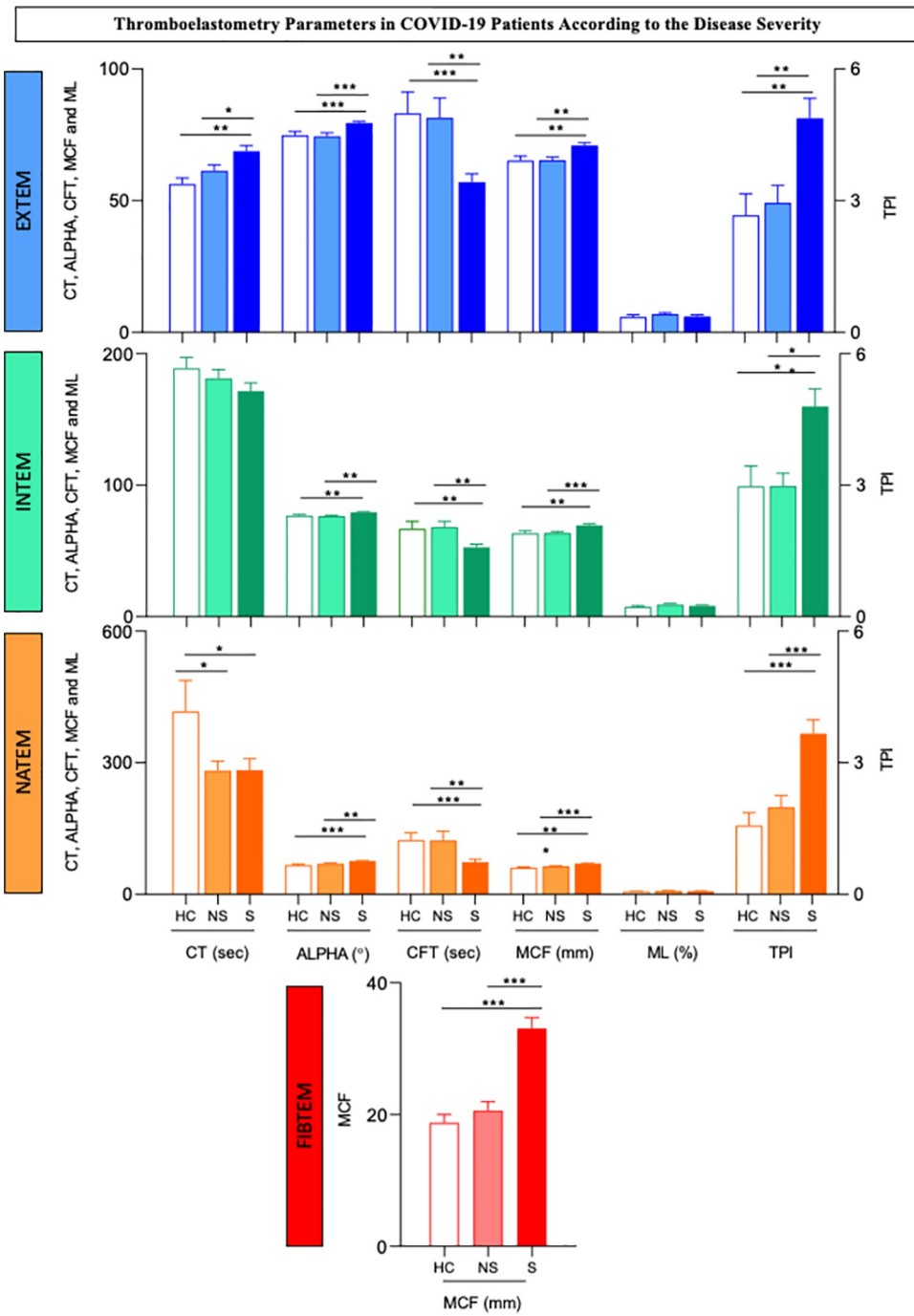

**Fig 3. Thromboelastometry parameters in COVID-19 patients according to disease severity.** Extrinsic (EXTEM) and intrinsic (INTEM) coagulation activity assays, nonactivated coagulation assays (NATEM) and functional assessments of fibrinogen assays (FIBTEM) were carried out as described in the Patients and Methods. The results for nonsevere (NS, n = 20, light color bars) and severe (S, n = 21, dark color bars) COVID-19 patients and healthy controls (n = 9, white bars) are presented as the mean values ± standard error. CT = clotting time, expressed in seconds; ALPHA = alpha angle, expressed in °; CFT = clot formation time, expressed in seconds; MCF = maximum clot firmness, expressed in mm; ML = maximum lysis, expressed in % and TPI = thrombodynamic potential index, calculated as [(100 x MCF)/(100-MCF)]/CFT. Significant differences are highlighted by connecting lines and *, ** or *** for p values ≤0.05, ≤0.01 and ≤0.001, respectively.

**Table 2. Performance of thromboelastometry parameters as complementary biomarkers to segregate COVID-19 patients.**

| PARAMETERS | DIAGNOSIS | | | | | PROGNOSIS | | | | |
|---|---|---|---|---|---|---|---|---|---|---|
| | HC x COVID | | | | | COVID (NS x S) | | | | |
| | AUC | Se (%) | Sp (%) | LR(-) | LR(+) | AUC | Se (%) | Sp (%) | LR(-) | LR(+) |
| EXTEM | | | | | | | | | | |
| CT | **0.7 (0.6–0.9)** | **46 (31–63)** | **100 (66–100)** | **0.5** | +∞ | 0.6 (0.5–0.8) | 86 (64–97) | 40 (19–64) | 0.4 | 1.4 |
| ALPHA | **0.7 (0.5–0.8)** | **51 (35–67)** | **89 (52–98)** | **0.6** | **4.6** | **0.8 (0.7–0.9)** | **81(58–94)** | **80 (56–94)** | **0.2** | **4.0** |
| CFT | **0.7 (0.6–0.8)** | **78 (49–80)** | **78 (40–97)** | **0.4** | **3.0** | **0.8 (0.6–0.9)** | **62 (39–82)** | **90 (68–99)** | **0.4** | **6.2** |
| MCF | **0.7 (0.5–0.8)** | **44 (29–60)** | **89 (52–98)** | **0.6** | **4.0** | **0.8 (0.6–0.9)** | **62 (39–82)** | **85 (62–97)** | **0.5** | **4.1** |
| ML | 0.6 (0.4–0.7) | 15 (6–29) | 100 (66–100) | 0.9 | +∞ | 0.6 (0.4–0.8) | 14 (3–36) | 100 (83–100) | 0.9 | +∞ |
| TPI | **0.7 (0.5–0.8)** | **61 (45–76)** | **78 (40–97)** | **0.5** | **2.7** | **0.8 (0.6–0.9)** | **62 (39–82)** | **90 (68–99)** | **0.4** | **6.2** |
| INTEM | | | | | | | | | | |
| CT | 0.6 (0.5–0.8) | 76 (60–88) | 56 (21–86) | 0.4 | 1.7 | 0.6 (0.4–0.8) | 62 (39–82) | 70 (46–88) | 0.5 | 2.1 |
| ALPHA | 0.6 (0.5–0.8) | 68 (52–82) | 67 (30–92) | 0.5 | 2.1 | **0.7 (0.6–0.9)** | **91 (70–99)** | **55 (32–77)** | **0.2** | **2.0** |
| CFT | 0.6 (0.5–0.8) | 71 (55–84) | 67 (30–92) | 0.4 | 2.1 | **0.8 (0.6–0.9)** | **91 (70–99)** | **55 (32–77)** | **0.2** | **2.0** |
| MCF | **0.7 (0.5–0.8)** | **76 (60–88)** | **56 (21–86)** | **0.4** | **1.7** | **0.8 (0.6–0.9)** | **67 (43–85)** | **80 (56–94)** | **0.4** | **3.3** |
| ML | 0.6 (0.4–0.7) | 22 (11–38) | 100 (66–100) | 0.8 | +∞ | 0.5 (0.4–0.7) | 95 (76–99) | 25 (9–49) | 0.2 | 1.3 |
| TPI | **0.7 (0.5–0.8)** | **73 (57–86)** | **67 (30–92)** | **0.4** | **2.2** | **0.8 (0.6–0.9)** | **71 (48–89)** | **70 (46–88)** | **0.4** | **2.4** |
| NATEM | | | | | | | | | | |
| CT | **0.8 (0.6–0.9)** | **56 (40–72)** | **89 (52–98)** | **0.5** | **5.1** | 0.5 (0.4–0.7) | 57 (34–78) | 60 (36–81) | 0.7 | 1.4 |
| AA | **0.8 (0.6–0.9)** | **42 (26–58)** | **100 (66–100)** | **0.6** | +∞ | **0.8 (0.6–0.9)** | **81 (58–94)** | **70 (46–88)** | **0.3** | **2.7** |
| CFT | **0.8 (0.6–0.9)** | **44 (29–60)** | **100 (66–100)** | **0.6** | +∞ | **0.8 (0.6–0.9)** | **91 (70–99)** | **60 (36–81)** | **0.2** | **2.3** |
| MCF | **0.8 (0.6–0.9)** | **76 (60–88)** | **67 (30–92)** | **0.4** | **2.3** | **0.8 (0.7–0.9)** | **86 (64–97)** | **65 (41–85)** | **0.2** | **2.5** |
| ML | 0.6 (0.4–0.7) | 15 (6–29) | 100 (66–100) | 0.9 | +∞ | 0.5 (0.3–0.7) | 86 (64–97) | 25 (9–49) | 0.6 | 1.1 |
| TPI | **0.8 (0.6–0.9)** | **47 (31–63)** | **100 (66–100)** | **0.5** | +∞ | **0.8 (0.7–0.9)** | **71 (48–89)** | **90 (68–99)** | **0.3** | **7.1** |
| FIBTEM | | | | | | | | | | |
| MCF | **0.8 (0.6–0.9)** | **68 (52–82)** | **89 (52–98)** | **0.4** | **6.2** | **0.9 (0.8–1.0)** | **76 (53–92)** | **90 (68–99)** | **0.3** | **7.6** |

HC = Healthy controls (n = 09); COVID = Patients with SARS-Cov-2 infection (n = 41); NS = Non-Severe COVID patients (n = 20); S = Severe COVID patients (n = 21); AUC = Area under the ROC curve; Se = Sensitivity; Sp = Specificity. Extrinsic (EXTEM) and Intrinsic (INTEM) coagulation activity assay, Non-Activated coagulation assay (NATEM) and functional assessment of Fibrinogen assay (FIBTEM). CT = clotting time, expressed in seconds; ALPHA = alpha angle, expressed in o; CFT = clot formation time, expressed in seconds; MCF = maximum clot firmness, expressed in mm; ML = maximum lysis, expressed in % and TPI = thrombodynamic potential index, calculated as: [(100 x MCF) / (100-MCF)] / CFT. *Cut-offs HC x COVID: EXTEM = CT≥66; ALPHA≥78; CFT≤70; MCF≥69; ML≥9; TPI≥2.9; INTEM = CT≤196; ALPHA≥77; CFT≤63; MCF≥63; ML≥11; TPI≥2.8; NATEM = CT≤255; ALPHA≥75; CFT≤72; MCF≥63; ML≥11; TPI≥2.9 and FIBTEM = MCF≥22. *Cut-offs NS x S: EXTEM = CT≥57; ALPHA≥78; CFT≤56; MCF≥70; ML≤2; TPI≥4; INTEM = CT≤175; ALPHA≥77; CFT≤62; MCF≥67; ML≤12; TPI≥3.4; NATEM = CT≤252; ALPHA≥74; CFT≤90; MCF≥64; ML≥4; TPI≥3.1 and FIBTEM = MCF≥27. Attributes with AUC >0.7 were highlighted by bold underline format and pre-selected for further analysis.

severe patients from HC with moderate/elevated global accuracy (AUC>0.7): CT, ALPHA angle, CFT, MCF and TPI from EXTEM; CT, ALPHA angle, CFT, MCF and TPI from INTEM; CT, ALPHA angle, CFT, MCF and TPI from NATEM and MCF from FIBTEM (Table 3). The preselected parameters underscored in Tables 2 and 3 were considered for further analysis.

## Decision tree algorithm proposed to classify COVID-19 patients according to disease severity

Considering the preselected thromboelastometry attributes with moderate/elevated global accuracy, decision tree algorithms were constructed to classify COVID-19 patients, and the

**Table 3. Performance of thromboelastometry parameters as complementary biomarkers to prognosis of SARS-Cov2 infection according to disease severity.**

| PARAMETERS* | PROGNOSIS | | | | | PROGNOSIS | | | | |
|---|---|---|---|---|---|---|---|---|---|---|
| | HC x NS | | | | | HC x S | | | | |
| | AUC | Se (%) | Sp (%) | LR(-) | LR(+) | AUC | Se (%) | Sp (%) | LR(-) | LR(+) |
| **EXTEM** | | | | | | | | | | |
| CT | **0.7 (0.5–0.8)** | **40 (19–64)** | **100 (66–100)** | **0.2** | +∞ | **0.8 (0.6–0.9)** | **52 (30–74)** | **100 (66–100)** | **0.5** | +∞ |
| ALPHA | 0.5 (0.3–0.7) | 10 (2–32) | 100 (66–100) | 0.9 | +∞ | **0.9 (0.7–1.0)** | **81 (58–94)** | **89 (52–98)** | **0.2** | **7.3** |
| CFT | 0.6 (0.4–0.8) | 55 (33–77) | 78 (40–97) | 0.6 | 2.5 | **0.8 (0.6–0.9)** | **90 (70–99)** | **67 (30–92)** | **0.1** | **2.7** |
| MCF | 0.5 (0.3–0.7) | 10 (2–32) | 100 (66–100) | 0.9 | +∞ | **0.8 (0.6–0.9)** | **67 (43–85)** | **89 (52–98)** | **0.4** | **6** |
| ML | 0.6 (0.4–0.8) | 20 (6–44) | 100 (66–100) | 0.8 | +∞ | 0.5 (0.3–0.7) | 57 (34–78) | 56 (21–86) | 0.8 | 1.3 |
| TPI | 0.6 (0.4–0.7) | 50 (27–73) | 78 (40–97) | 0.6 | 2.3 | **0.8 (0.6–0.9)** | **67 (43–85)** | **89 (52–98)** | **0.4** | **6** |
| **INTEM** | | | | | | | | | | |
| CT | 0.6 (0.4–0.8) | 70 (46–88) | 56 (21–86) | 0.2 | +∞ | **0.7 (0.5–0.8)** | **81 (58–94)** | **56 (21–86)** | **0.3** | **1.8** |
| ALPHA | 0.5 (0.3–0.7) | 85 (62–97) | 33 (8–70) | 0.5 | 1.3 | **0.8 (0.6–0.9)** | **90 (70–99)** | **67 (30–92)** | **0.1** | **2.7** |
| CFT | 0.5 (0.3–0.7) | 85 (62–97) | 33 (8–70) | 0.5 | 1.3 | **0.8 (0.6–0.9)** | **90 (70–99)** | **67 (30–92)** | **0.1** | **2.7** |
| MCF | 0.5 (0.3–0.7) | 45 (23–68) | 67 (30–92) | 0.8 | 1.4 | **0.8 (0.6–0.9)** | **95 (76–99)** | **56 (21–86)** | **0.1** | **2.1** |
| ML | 0.6 (0.4–0.8) | 25 (9–49) | 100 (66–100) | 0.8 | +∞ | 0.5 (0.4–0.7) | 19 (6–42) | 100 (66–100) | 0.8 | +∞ |
| TPI | 0.5 (0.3–0.7) | 55 (32–77) | 67 (30–92) | 0.7 | 1.7 | **0.8 (0.6–0.9)** | **90 (70–99)** | **67 (30–92)** | **0.1** | **2.7** |
| **NATEM** | | | | | | | | | | |
| CT | **0.8 (0.6–0.9)** | **40 (19–64)** | **100 (66–100)** | **0.6** | +∞ | **0.8 (0.6–0.9)** | **48 (26–70)** | **100 (66–100)** | **0.5** | +∞ |
| ALPHA | 0.6 (0.4–0.8) | 95 (75–99) | 33 (8–70) | 0.2 | 1.4 | **0.9 (0.7–1.0)** | **81 (58–94)** | **78 (40–97)** | **0.2** | **3.6** |
| CFT | 0.6 (0.4–0.8) | 85 (62–97) | 44 (14–79) | 0.3 | 1.5 | **0.9 (0.7–1.0)** | **62 (39–82)** | **100 (66–100)** | **0.4** | +∞ |
| MCF | **0.7 (0.5–0.8)** | **85 (62–97)** | **44 (14–79)** | **0.3** | **1.5** | **0.9 (0.7–1.0)** | **86 (64–97)** | **78 (40–97)** | **0.2** | **3.9** |
| ML | 0.6 (0.4–0.8) | 20 (6–44) | 100 (66–100) | 0.2 | +∞ | 0.5 (0.4–0.7) | 86 (64–97) | 33 (8–70) | 0.4 | 1.3 |
| TPI | 0.6 (0.5–0.8) | 85 (62–97) | 56 (21–86) | 0.3 | 1.9 | **0.9 (0.7–1.0)** | **71 (48–89)** | **100 (66–100)** | **0.3** | +∞ |
| **FIBTEM** | | | | | | | | | | |
| MCF | 0.6 (0.4–0.8) | 40 (19–64) | 89 (52–98) | 0.7 | 3.6 | **1.0 (0.8–1.0)** | **95 (76–99)** | **89 (52–98)** | **0.1** | **8.6** |

HC = Healthy controls (n = 09); COVID = Patients with SARS-Cov-2 infection (n = 41); NS = Non-Severe COVID patients (n = 20); S = Severe COVID patients (n = 21); AUC = Area under the ROC curve; Se = Sensitivity; Sp = Specificity. Extrinsic (EXTEM) and Intrinsic (INTEM) coagulation activity assay, Non-Activated coagulation assay (NATEM) and functional assessment of Fibrinogen assay (FIBTEM). CT = clotting time, expressed in seconds; ALPHA = alpha angle, expressed in o; CFT = clot formation time, expressed in seconds; MCF = maximum clot firmness, expressed in mm; ML = maximum lysis, expressed in % and TPI = thrombodynamic potential index, calculated as: [(100 x MCF) / (100-MCF)] / CFT. *Cut-offs HC x NS: EXTEM = CT≥66; ALPHA≤65; CFT≥70; MCF≤53; ML≥9; TPI≥2.9; INTEM = CT≤191; ALPHA≤79; CFT≥49; MCF≤61; ML≥11; TPI≥2.8; NATEM = CT≤223; ALPHA≥61; CFT≤130; MCF≥59; ML≥11; TPI≥1.1 and FIBTEM = MCF≥22. *Cut-offs HC x S: EXTEM = CT≥66; ALPHA≥78; CFT≤72; MCF≥69; ML≤6; TPI≥3.7; INTEM = CT≤196; ALPHA≥77; CFT≤62; MCF≥63; ML≥11; TPI≥2.8; NATEM = CT≤240; ALPHA≥74; CFT<72; MCF≥64; ML≥4; TPI≥2.9 and FIBTEM = MCF≥22. Attributes with AUC >0.7 were highlighted by bold underline format and pre-selected for further analyses.

data are presented in Fig 4. A data analysis was carried out to identify root and branch attributes to classify patients with higher accuracy as follows: HC *vs* COVID-19 (Fig 4A), HC *vs* NS (Fig 4B), HC *vs* S (Fig 4C) and NS *vs* S (Fig 4D). The decision trees were constructed using the cutoff values defined by the ROC curve analysis (Tables 2 and 3). The decision tree algorithm for HC *vs* COVID classification proposed the use of FIBTEM-MCF (22 mm) and EXTEM-CT (66 seconds) as the root and first branch attributes, respectively, to yield elevated accuracy (80%, LOOCV = 80%) (Fig 4A). Classification of HC *vs* NS with elevated accuracy (83%, LOOCV = 83%) was obtained by using EXTEM-CT (66 seconds) and NATEM-CT (223 seconds) as the root and first branch attributes, respectively (Fig 4B). Differentiation of S from HC was achieved with high accuracy (93%, LOOCV = 93%) using FIBTEM-MCF (22 mm) as a single root attribute (Fig 4C). Additionally, differentiation between NS and S COVID-19

**Proposal of Decision Tree Algorithm for Thromboelastometry Parameters to Differentiate COVID-19 Patients According to Disease Severity**

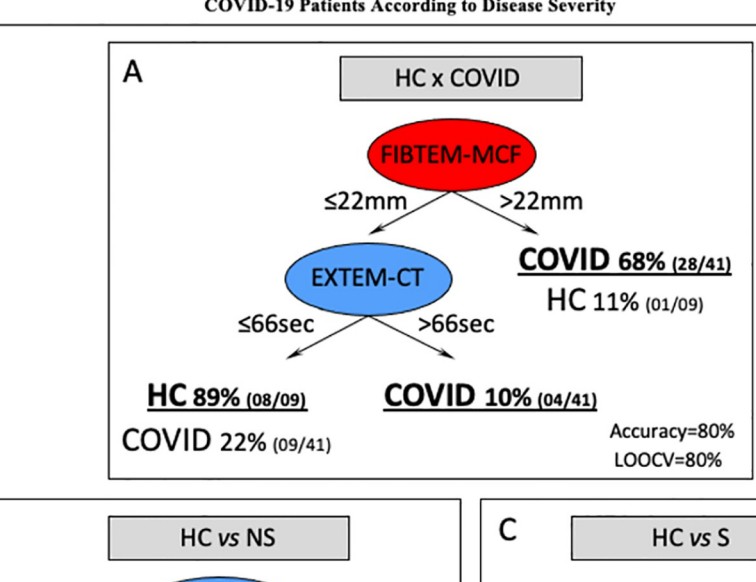

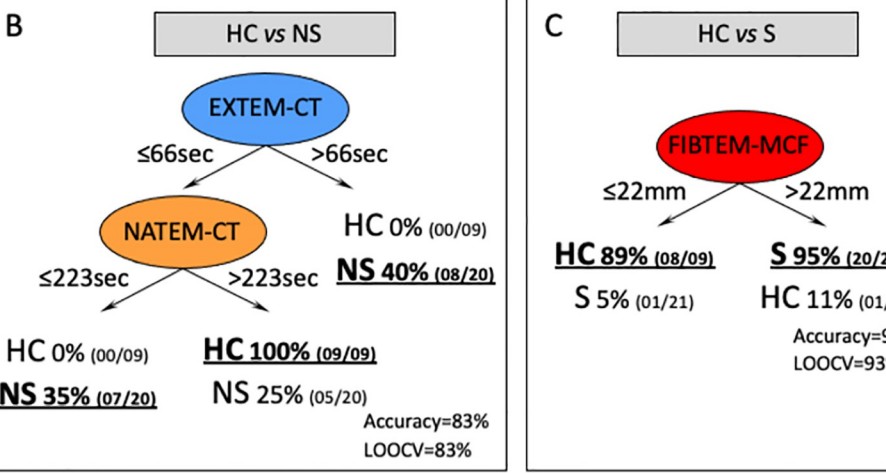

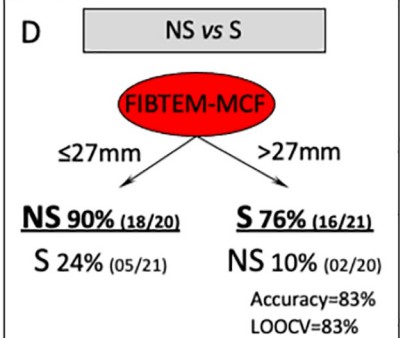

**Fig 4. Proposed decision tree algorithm for thromboelastometry parameters to differentiate COVID-19 patients according to disease severity.** Decision tree algorithm using thromboelastometric parameters was generated to classify: (A) COVID-19 patients from healthy controls (HC x COVID); (B) nonsevere patients from healthy controls (HC x NS); (C) severe patients from healthy controls (HC x S); and (D) severe patients from nonsevere COVID-19 patients (NS x S). The numbers at branches indicate the cutoff values used to classify each group. The global accuracy and leave-one-out-cross-validation (LOOCV) scores are provided in the figure.

patients was possible by applying a simple algorithm based on the use of FIBTEM-MCF (27 mm) with elevated accuracy (83%, LOOCV = 83%) (Fig 4D).

## Stepwise and one-step analysis to classify COVID-19 patients

The proposed decision tree algorithms for classifying subgroups of COVID-19 patients and healthy controls were further presented as stepwise and one-step scatter plot analyses, as shown in Fig 5. The stepwise analysis to classify HC *vs* COVID-19 comprises two consecutive analyses, including FIBTEM-MCF followed by EXTEM-CT. In the first round of analysis, it was possible to precisely classify 28 out of 41 COVID-19 patients with one misclassification of HC as COVID-19. Samples with FIBTEM-MCF below or equal to 22 mm were further analyzed for the EXTEM-CT profile. The results of EXTEM-CT below or equal to 66 seconds classified 08/08 as HC and misclassified 09/13 COVID patients (Fig 5A). Overall, the stepwise analysis correctly classified 40 out of 50 subjects.

The stepwise algorithm for classifying HC *vs* NS consists of EXTEM-CT followed by NATEM-CT. In the first step, the decision tree accurately classified 08 out of 20 nonsevere COVID-19 patients with no misclassification. Samples with EXTEM-CT below or equal to 66 seconds were moved forward to the NATEM-CT analysis. NATEM-CT results below or equal to 223 seconds were used to classify 07/12 as NS, with no misclassification. Samples with NATEM-CT higher than 223 seconds were categorized as HC, with 5 misclassifications (Fig 5B). The final analysis correctly classified 24 out of 29 subjects.

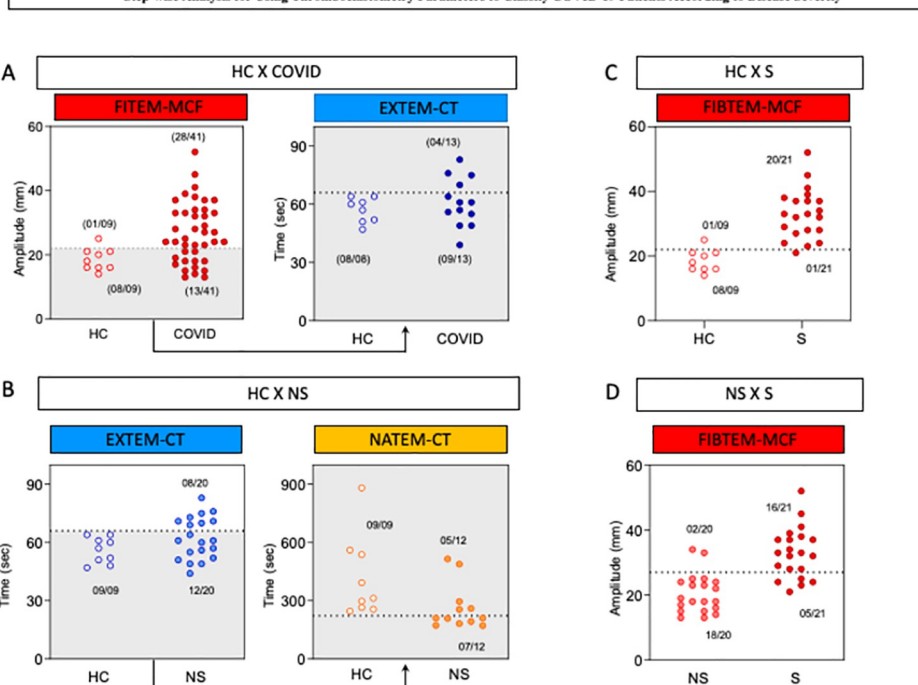

**Fig 5. Stepwise analysis for using thromboelastometry parameters to classify COVID-19 patients according to disease severity.** Scatter plot distributions illustrate the stepwise and one-step analysis proposed to classify: (A) COVID-19 patients from healthy controls (HC x COVID); (B) nonsevere patients from healthy controls (HC x NS); (C) severe patients from healthy controls (HC x S); and (D) severe patients from nonsevere COVID-19 patients (NS x S). The dotted lines represent the cutoff values selected by ROC curve analysis with the highest global accuracy (AUC). In the stepwise analysis, the gray background underscored the samples used in the second round of analysis. The proportions of accurate classifications and T are provided in the figure.

The algorithm proposed to classify HC *vs* S based on the single use of FIBTEM-MCF. The classification tree accurately classified samples with FIBTEM-MCF higher than 22 mm as severe COVID-19 patients, with 20 out of 21 hits and one misclassification of HC as severe COVID-19 (Fig 5C).

A one-step analysis was also proposed to differentiate S from NS COVID patients using FIBTEM-MCF as a single parameter. The algorithm precisely classified 16 out of 21 severe COVID-19 patients with FIBTEM-MCF higher than 27 mm, with only two misclassifications of NS as severe COVID-19 (Fig 5D).

## Discussion

The assessment of coagulation derangement is not an easy task, especially in the prothrombotic pole. In this regard, viscoelastic tests have emerged as a promising technique to enable the detailed analysis of all coagulation stages, including initiation (CT), fibrin polymerization (ALPHA and CFT), fibrinogen and platelet contributions (MCF) and fibrinolysis (ML) [7, 8, 11, 12]. Few studies have characterized the coagulopathic process in COVID-19 patients, and of these studies, most are performed in critically ill patients. Knowledge of the specific alterations of thromboelastometric parameters in nonsevere and severe COVID-19 patients may allow for the future characterization of biomarkers useful for thromboprophylaxis or decision making in patients with low-risk of thromboembolic events.

In the present work, we characterized the thromboelastometric profile of nonsevere and severe forms of COVID-19 compared to healthy controls. Our findings showed a clear hypercoagulant profile in severe forms of COVID-19 based on the evaluation of EXTEM parameters, as previously described [7, 16]. A single parameter (EXTEM-CT) did not fulfill the hypercoagulability diagnosis, with values in the normal range [14]. However, all INTEM parameters, even CT, which is known to be more heparin-sensitive, support a clear hypercoagulant pattern in severe COVID-19 [26]. Since heparin thromboprophylaxis was administered to 90% of severe patients included in the present investigation and they still presented a clear INTEM hypercoagulability profile, we hypothesize that heparin therapy in usual doses may not have been sufficient to control the thrombotic tendency in severe COVID-19 patients and that INTEM analysis may represent a relevant biomarker to predict and adjust thromboprophylaxis management in COVID-19 patients, even knowing that INTEM is not highly sensitive to LMWH [25–27]. Further studies with a larger number of patients are necessary to confirm the applicability of INTEM-CT as a biomarker for insufficient anticoagulation therapy.

Previous evaluations of NATEM parameters have not been performed in COVID-19 patients. Altered NATEM profiles have been well described in patients with bacterial sepsis [17–19]. In septic patients, the induction of tissue factor (TF) expression in circulating mononuclear cells may explain the occurrence of disseminated thrombi and multiple organ failure [17–19]. As no specific activator is added in the NATEM assay, the production of thrombin will be triggered by pre-existing circulating TF. Our results showed that both severe and nonsevere COVID-19 patients presented altered NATEM profiles, which were characterized by shortening of CT. As the nonsevere patients exhibited this thromboelastometric parameter (NATEM-CT) suggestive of incipient hypercoagulopathy, our results suggested that even patients not hospitalized with nonsevere forms of COVID-19 may present a risk of developing thromboembolic events, which is probably due to increased levels of circulating TF expression. Additional studies of nonsevere patients may confirm whether altered NATEM-CT would represent a thromboelastometric parameter useful as a predictor of worse clinical evolution, thereby allowing for earlier intervention to avoid evolution to more severe clinical forms.

Moreover, as NATEM is more sensitive to LMWH than INTEM, it is possible to postulate that this parameter could be used to manage anticoagulation dosing [27]. To the best of our knowledge, this is the first study in the literature to show endogenous coagulation activation using the NATEM test in COVID-19 nonsevere and severe forms and to describe the coagulation derangement of nonsevere patients.

The analysis of the FIBTEM results reinforces the abovementioned hypercoagulant findings observed for severe COVID-19 patients. Moreover, the FIBTEM-MCF parameter was able to differentiate nonsevere and severe patients. Our data corroborate previous reports in which higher FIBTEM-MCF results were found for severely ill COVID-19 patients than healthy controls and patients in regular wards [14]. As fibrinogen is an acute phase protein that is progressively elevated, this finding is further supported by previous data that severe patients exhibited higher fibrinogen levels than nonsevere patients [4, 28].

The TPI is calculated from the CFT and MCF thromboelastometric parameters and represents a robust measure of thrombogenic potential [8, 10]. In our study, TPI was able to differentiate severe from nonsevere patients and healthy controls in the EXTEM, INTEM and NATEM assays. Multiple clinical trials have evaluated thromboinflammatory biomarkers associated with poor prognosis among COVID-19 patients. To our knowledge, previous reports have not evaluated the TPI parameter in COVID-19 patients. Our findings of a hypercoagulable profile measured by TPI represent novel insights for the application of biomarkers in personalized antithrombotic therapy for COVID-19 [29].

Fibrinolysis shutdown has been hypothesized to occur during COVID-19 progression [30–32], and a local pulmonary hyperfibrinolysis process may also occur in COVID-19 patients [33]. In general, fibrinolysis potential has been associated with D-dimer levels. However, the increased D-dimer levels observed in COVID-19 patients are not necessarily associated with a higher fibrinolysis profile, as only about 0.02 to 0.2% of the fibrinogen mass is cleaved [14]. Our data did not demonstrate any significant differences in ML between severe and nonsevere COVID-19 patients or relative to the healthy controls.

To date, thromboelastometry cutoff values for defining hypercoagulability remain controversial. In the present study, we presented a decision tree algorithm based on cutoff values derived from a ROC curve analysis to classify COVID-19 patients according to clinical status. The proposed algorithm uses a stepwise or single-step approach and represents an objective tool for application as a complementary laboratory method to classify COVID-19 patients according to disease severity. Additionally, the same principles of the decision tree could be used to help attending physicians identify patients requiring further enhancement of antithrombotic prophylaxis.

As per current critical care guidelines, most of our patients were receiving some form of heparin during blood collection. Thus, some of our results could have been influenced by its anticoagulant properties, nevertheless thromboelastometric tests are less affected than other conventional coagulation tests. In addition, our reduced sample size prevents us from assessing coagulability differences in the broad spectrum of COVID-19 severity. Further studies are necessary to get into detail of the specific coagulability status presented by severe patients with different levels of organ dysfunctions.

Altogether, our findings demonstrate that patients with severe COVID-19 exhibited a thromboelastometry profile with clear hypercoagulability dysfunction, and it was significantly different from the profiles of nonsevere COVID-19 patients and healthy controls. Of note, the FIB-MCF profile was selected as a putative biomarker to differentiate patients with severe COVID-19 from nonsevere patients and healthy controls, and it presented moderate/elevated accuracy. The results showed that the TPI data analysis from EXTEM, INTEM and NATEM were significantly different in nonsevere forms compared to severe clinical forms.

Additionally, the NATEM data suggested that nonsevere and severe COVID-19 patients presented endogenous coagulation activation (reduced CT and CFT times).

## Conclusion

Our work demonstrated in our population, that thromboelastometry is useful to detect endogenous coagulation activation in both severe and nonsevere COVID-19 patients (reduced CT and CFT in NATEM). That TPI is enhanced in severe patients and that selected thromboelastometric parameters may be used to separate severe from nonsevere patients with moderate/high accuracy.

## Supporting information

**S1 Table. Detailed thromboelastometric results.** EXTEM = Extrinsic coagulation activity assay; INTEM = Intrinsic coagulation activity assay; NATEM = Non-Activated coagulation assay and FIBTEM = functional assessment of Fibrinogen assay. CT (expressed in seconds) = clotting time: refers to thrombin formation, timeframe from activation until an amplitude of 2mm; ALPHA (expressed in ˚) = defined as the angle between the middle axis and the tangent to the clotting curve through the 2mm amplitude point; CFT (expressed in seconds) = clot formation time: refers to the dynamic formation of fibrin, timeframe between 2 mm and 20 mm of clot amplitude; MCF (expressed in mm) = maximum clot firmness: refers to the maximum firmness of the clot, proportional to the amount of fibrinogen and platelets; ML (expressed in %) = maximum lysis: represents the percentage of clot reduction after initiation of fibrinolysis. § Data are expressed as Mean±SD. † P values were calculated with independent-samples Student's t-tests for continuous variables. Significant differences are underscored by letters "a", "b" and "c" for pairwise comparisons between NS vs HC, S vs HC and S vs NS, respectively. (DOCX)

## Acknowledgments

The study was supported by the Hospital Universitário de Brasília (HUB-EBSERH) from Universidade de Brasília (UnB) and Hospital Regional da Asa Norte (HRAN). The authors acknowledge the DASA Laboratory for performing the ROTEM analysis and Werfen Medical for supplying reagents for the ROTEM analysis. R.B.A. is enrolled in the Programa de Pós-graduação em Ciências Médicas da UnB.

## Author Contributions

**Conceptualization:** Rodrigo B. Aires, Laurence R. do Amaral, Matheus de S. Gomes, Otávio T. Nóbrega, Ciro M. Gomes, Patricia S. Kurizky, Cleandro P. Albuquerque, Olindo A. Martins-Filho, Licia Maria H. da Mota.

**Data curation:** Rodrigo B. Aires, Heidi Luise Schulte, Patricia S. Kurizky, Olindo A. Martins-Filho.

**Formal analysis:** Rodrigo B. Aires, Alexandre A. de S. M. Soares, Ana Paula M. Gomides, André M. Nicola, Dayde Lane M. da Silva, Flávia D. Xavier, Isabelle S. Luz, Laila S. Espindola, Laurence R. do Amaral, Matheus de S. Gomes, Otávio T. Nóbrega, Wagner Fontes, Ciro M. Gomes, Patricia S. Kurizky, Cleandro P. Albuquerque, Olindo A. Martins-Filho, Licia Maria H. da Mota.

**Funding acquisition:** Laila S. Espindola, Olindo A. Martins-Filho.

**Investigation:** André M. Nicola, Dayde Lane M. da Silva, Francielle P. Martins, Gabriela P. J. Santos, Isabelle S. Luz, Liza F. Felicori, Luciana A. Naves, Maíra R. M. de Carvalho, Ciro M. Gomes, Olindo A. Martins-Filho, Licia Maria H. da Mota.

**Methodology:** Rodrigo B. Aires, Andréa Teixeira-Carvalho, Flávia D. Xavier, Heidi Luise Schulte, Laurence R. do Amaral, Liza F. Felicori, Otávio T. Nóbrega, Patrícia Albuquerque, Wagner Fontes, Ciro M. Gomes, Patricia S. Kurizky, Cleandro P. Albuquerque, Olindo A. Martins-Filho, Licia Maria H. da Mota.

**Project administration:** Rodrigo B. Aires, Licia Maria H. da Mota.

**Resources:** Francielle P. Martins, Gabriela P. J. Santos, Laila S. Espindola, Luciana A. Naves, Maíra R. M. de Carvalho, Ciro M. Gomes, Patricia S. Kurizky, Olindo A. Martins-Filho, Licia Maria H. da Mota.

**Software:** Laurence R. do Amaral, Matheus de S. Gomes, Cleandro P. Albuquerque, Olindo A. Martins-Filho.

**Supervision:** Patricia S. Kurizky, Olindo A. Martins-Filho, Licia Maria H. da Mota.

**Validation:** Patrícia Albuquerque, Patricia S. Kurizky, Cleandro P. Albuquerque, Olindo A. Martins-Filho, Licia Maria H. da Mota.

**Visualization:** Eliana T. de Gois, Heidi Luise Schulte, Patrícia Albuquerque, Wagner Fontes, Patricia S. Kurizky, Cleandro P. Albuquerque, Olindo A. Martins-Filho, Licia Maria H. da Mota.

**Writing – original draft:** Rodrigo B. Aires, Otávio T. Nóbrega, Ciro M. Gomes, Patricia S. Kurizky, Cleandro P. Albuquerque, Olindo A. Martins-Filho, Licia Maria H. da Mota.

**Writing – review & editing:** Rodrigo B. Aires, Alexandre A. de S. M. Soares, Ana Paula M. Gomides, André M. Nicola, Andréa Teixeira-Carvalho, Dayde Lane M. da Silva, Eliana T. de Gois, Flávia D. Xavier, Francielle P. Martins, Gabriela P. J. Santos, Heidi Luise Schulte, Isabelle S. Luz, Laila S. Espindola, Liza F. Felicori, Luciana A. Naves, Maíra R. M. de Carvalho, Matheus de S. Gomes, Otávio T. Nóbrega, Patrícia Albuquerque, Wagner Fontes, Ciro M. Gomes, Patricia S. Kurizky, Cleandro P. Albuquerque, Olindo A. Martins-Filho, Licia Maria H. da Mota.

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
