## [Decision Letter · Decision Letter 0]

21 Jul 2021

PONE-D-21-17952

Thromboelastometry demonstrates endogenous coagulation activation in nonsevere and severe COVID-19 patients and has applicability as a decision algorithm for intervention

PLOS ONE

Dear Dr. aires,

Thank you for submitting your manuscript to PLOS ONE. After careful consideration, we feel that it has merit but does not fully meet PLOS ONE’s publication criteria as it currently stands. Therefore, we invite you to submit a revised version of the manuscript that addresses the points raised during the review process.

We look forward to receiving your revised manuscript.

Kind regards,

Andrea Ballotta

Academic Editor

PLOS ONE

Journal Requirements:

2. In the Methods, please state:

- Why written consent could not be obtained

- Whether the Institutional Review Board (IRB) approved use of digital consent

For more information, please see our guidelines for human subjects research: https://journals.plos.org/plosone/s/submission-guidelines#loc-human-subjects-research

“The study was supported by Conselho Nacional de Desenvolvimento Científico e 557 Tecnológico (CNPq), Coordenação de Aperfeiçoamento de Pessoal de Nível Superior 558 (CAPES), the Hospital Universitário de Brasília (HUB-EBSERH) from Universidade de 559 Brasília (UnB) and Hospital Regional da Asa Norte (HRAN).”

 “The authors are grateful for the financial support provided by Conselho Nacional de Desenvolvimento Científico e Tecnológico (CNPq) and Coordenação de Aperfeiçoamento de Pessoal de Nível Superior (CAPES). LAN, MSG, OTN, WF and OAMF thank CNPq for the PQ fellowship program. The funders had no role in the study design, data collection and analysis, decision to publish, or preparation of the manuscript.”

6. Please amend the manuscript submission data (via Edit Submission) to include author Alexandre A. de S. M. Soares, Ana Paula M. Gomides, André M. Nicola, Andrea Teixeira-Carvalho, Dayde Lane M. da Silva, Eliana T. de Gois, Flávia D. Xavier, Francielle P. Martins, Gabriela P. J. Santos, Heidi Luise Schulte, Isabelle S. Luz, Laila S. Espindola, Laurence R. do Amaral, Liza F. Felicori, Luciana A. Naves, Maíra R. M. de Carvalho, Matheus de S. Gomes, Otávio T. Nóbrega, Patrícia Albuquerque, Wagner Fontes, Ciro M. Gomes, Patricia S. Kurizky1, Cleandro P. Albuquerque, Olindo A. Martins-Filho and Licia Maria H. da Mota.

Additional Editor Comments:

The paper needs major revision

Reviewers' comments:

Reviewer's Responses to Questions

**Comments to the Author**

1. Is the manuscript technically sound, and do the data support the conclusions?

Reviewer #1: Yes

2. Has the statistical analysis been performed appropriately and rigorously? 

Reviewer #1: I Don't Know

3. Have the authors made all data underlying the findings in their manuscript fully available?

Reviewer #1: No

4. Is the manuscript presented in an intelligible fashion and written in standard English?

Reviewer #1: Yes

5. Review Comments to the Author

Reviewer #1: In this work, the authors aimed to give their contribution to the assessment of the viscoelastic methods, in particular thromboelastometry ROTEM, in the COVID-19 setting with the addition of a statistical model for the stratification of the disease severity based on selected thromboelastometric values. The employment of the NATEM test in the study evaluation represents the very novelty point in comparison to what have been done previously by the other research groups.

My major issue concerning this study is the initial stratification of the patients operated by the authors. In the previous published research regarding the topic, it was cleared that the most important differences for thromboelastometric evaluation were present between the critical patients (admitted to ICU, on mechanical ventilation) vs. non critical subjects (that could be patients admitted to regular ward or on home therapy, but more frequently hospitalized). When critically ill patients were analyzed together with the non-critical ones the differences were smaller or became non-significant. In the light of this, I wander how the series of the authors would react to a further stratification of the Severe group into critical (points 4 and 5 of the criteria) and non-critical. Moreover, it may be interesting to add platelet count to the decision tree algorythm because it's another important parameter contributing to the prothrombotic evolution in the COVID-19 positive patients.

6. PLOS authors have the option to publish the peer review history of their article (what does this mean?). If published, this will include your full peer review and any attached files.

Reviewer #1: No

---

## [Author Response · Author response to Decision Letter 0]

15 Aug 2021

Dear editors

 This document is intended to provide responses to each point addressed by the Journal and Reviewers.

 The authors would like to thank for all the suggestions made, including the inclusion of our protocol in the protocol repository: protocols.io (dx.doi.org/10.17504/protocols.io.bwvbpe2n)

Journal Requirements

1. Re-checked PLOS ONE´s style requirements.

 We found some situations where the Journal´s style requirements were not met. All of those were corrected.

2. We decided to avoid Written Consent as the risk of manipulating objects between healthy and sick individuals could, theoretically, increase the viral transmission in the institution.

 So, the use of digital consent was requested and approved by our ethic review board, namely CONEP (Comissão Nacional de Ética em Pesquisa).

 Both informations were included in the manuscript, in the Methods section.

3. Amended the information included in the Funding Statement so it is the same mentioned in the Financial Disclosure. 

4. The new text with Funding Statement was included at the end of the Cover Letter, as requested.

5. To a complete understanding of the results and conclusions made in the paper, all data were made available: not only in graphic visualization, but also in a Supporting Table with all averages, standard deviation and p values. 

 As suggested, we also uploaded all raw data to a public repository, namely Harvard Dataverse: https://doi.org/10.7910/DVN/NWSVZA

6. All authors were included in the Manuscript Submission Data.

Reviewer responses to Questions

1. Reviewer #1 answered positively; nothing to change.

 2. Our statistical analysis was planned by three senior authors. Author MSG and LRA are both mathematicians at UFU (Federal University of Uberlandia) and actually work with the development of machine learning algorithms. Besides them, OAMF is a senior researcher with great expertise in bio-statistics and mathematic models of generalizability. They have been working together planning and participating in many studies involving the necessity of advanced mathematic evaluations. After that, other two authors, CPA and OTN, two researchers specially involved with statistical analysis, validated and re-tested all the data obtained in this study.

 Besides that, all our raw data and statistical analysis are available in the public repository Harvard Dataverse (DOI included in the Supporting information): https://doi.org/10.7910/DVN/NWSVZA

3. All our raw data are public and there isn´t any objection to make it available. For the understanding of the results and conclusions made in the paper, the data included are enough, as we included not only the p value, but also, means and standard deviation. We understand that it may be more difficult to visualize the numbers just in the graphics, but in the S1 Table, all the numbers are available and may easily be accessed and analyzed. 

 If it is not enough, it will be a pleasure for us to provide access to all raw data (Public repository Harvard Dataverse): https://doi.org/10.7910/DVN/NWSVZA

4. Reviewer #1 answered positively; nothing to change.

5. 

 Thank you very much for the suggestion.

At the time of the research we were faced with the necessity of stratifying our patients. At that time, a used guideline available was published by the WHO and separated individuals using the criteria selected in this study. As the time passed, possibily we would use other criteria, but at that time, it was a reasonable one.

Besides the novel use of NATEM in general COVID patients, other important topic in our study was the finding that even nonsevere patients already have hemostatic alterations, especially in NATEM. Moreover, the original description of TPI (thrombodynamic index) is other strong fact in our study. This parameter, as NATEM CT, also maintained the alterations described even when not separating the COVID-19 patients.

Thinking in this perspective, even with more strict criteria it probably wouldn’t modify the results, as both groups, nonsevere and severe, showed endogenous activation, represented by shorter CT in NATEM. Moreover, TPI probably wouldn´t also be altered by more strict criteria for classifying patients as severe. 

We respectfully consider that even changing the classification criteria, our findings justify the publication and highlight more aspects of this challenging disease. After all, we have commented in the discussion that our sample size wouldn’t be able to detect different severities in the severe group.

 The suggestion of adding platelet count is very good also; unfortunately, it was not available in most of our patients. As these patients are from public service hospitals, even simple exams may be unavailable at some periods. That was the case during the inclusion period: platelets, d-dimers and even creatinine were unavailable and so, not considered for the mathematic models.

6. Reviewer #1 answered positively; nothing to change.

 We hope that our answers are appropriate, but we are completely available to explain any other detail or doubt that may appear.

---

## [Decision Letter · Decision Letter 1]

31 Dec 2021

Thromboelastometry demonstrates endogenous coagulation activation in nonsevere and severe COVID-19 patients and has applicability as a decision algorithm for intervention

PONE-D-21-17952R1

Dear Dr. aires,

We’re pleased to inform you that your manuscript has been judged scientifically suitable for publication and will be formally accepted for publication once it meets all outstanding technical requirements.

Kind regards,

Andrea Ballotta

Academic Editor

PLOS ONE

Additional Editor Comments (optional):

On the basis of the reviewer's evaluation after your response and my judgement i deem the paper "Thromboelastometry demonstrates endogenous coagulation activation in nonsevere and severe COVID-19 patients and has applicability as a decision algorithm for intervention" suitable for publication

Reviewers' comments:

Reviewer's Responses to Questions

**Comments to the Author**

1. If the authors have adequately addressed your comments raised in a previous round of review and you feel that this manuscript is now acceptable for publication, you may indicate that here to bypass the “Comments to the Author” section, enter your conflict of interest statement in the “Confidential to Editor” section, and submit your "Accept" recommendation.

Reviewer #1: All comments have been addressed

2. Is the manuscript technically sound, and do the data support the conclusions?

Reviewer #1: Yes

3. Has the statistical analysis been performed appropriately and rigorously? 

Reviewer #1: I Don't Know

4. Have the authors made all data underlying the findings in their manuscript fully available?

Reviewer #1: Yes

5. Is the manuscript presented in an intelligible fashion and written in standard English?

Reviewer #1: Yes

6. Review Comments to the Author

Reviewer #1: All my observations and suggestions have been addressed thourougly. I have no further issues with this submission.

7. PLOS authors have the option to publish the peer review history of their article (what does this mean?). If published, this will include your full peer review and any attached files.

Reviewer #1: No

---

## [Editor Report · Acceptance letter]

6 Jan 2022

PONE-D-21-17952R1 

Thromboelastometry demonstrates endogenous coagulation activation in nonsevere and severe COVID-19 patients and has applicability as a decision algorithm for intervention 

Dear Dr. Aires:

I'm pleased to inform you that your manuscript has been deemed suitable for publication in PLOS ONE. Congratulations! Your manuscript is now with our production department. 

Kind regards, 

on behalf of

Dr. Andrea Ballotta 

Academic Editor

PLOS ONE